# OUT-OF-DISTRIBUTION DETECTION IN FEW-SHOT CLASSIFICATION

## ABSTRACT

In many real-world settings, a learning model must perform few-shot classification: learn to classify examples from unseen classes using only a few labeled examples per class. Additionally, to be safely deployed, it should have the ability to detect out-of-distribution inputs: examples that do not belong to any of the classes. While both few-shot classification and out-of-distribution detection are popular topics, their combination has not been studied. In this work, we propose tasks for out-of-distribution detection in the few-shot setting and establish benchmark datasets, based on four popular few-shot classification datasets. Then, we propose two new methods for this task and investigate their performance.

In sum, we establish baseline out-of-distribution detection results using standard metrics on new benchmark datasets and show improved results with our proposed methods.

## 1 INTRODUCTION

Few-shot learning, at a high-level, is the paradigm of learning where a model is asked to learn about new concepts from only a few examples (Fei-Fei et al., 2006; Lake et al., 2015). In the case of few-shot classification, a model must classify examples from novel classes, based on only a few labelled examples from each class. The model has to quickly learn (or adapt) a classifier given this very limited amount of learning signal. This paradigm of learning is attractive for the fundamental reason that it resembles how an intelligent system in the real-world has to behave. Unlike the traditional supervised setting, in most real-world settings we would not have access to millions of labelled examples, but would benefit if a few-shot classifier could be deployed, for example, to recognize the facial gestures of a new user, in order to improve human-computer interaction for individuals with motor disabilities (Wang et al., 2019).

For an intelligent system to be deployed in the real-world, not only does it have to do well on the designated task, but perhaps more importantly it should defer its actions when faced with unforeseen situations. In particular, when an input is invalid, or does not belong to any of the target classes, the system should identify the input as out-of-distribution. Successfully detecting out-of-distribution examples is crucial in a safety critical environment. In the supervised setting, out-of-distribution detection has been studied from many different angles (Hendrycks & Gimpel, 2016; Nalisnick et al., 2018), but this task has not been investigated in the few-shot setting.

Worryingly, the current state-of-the-art learning systems, deep neural networks, are known to be unreasonably confident about inputs unrecognizable to humans (Nguyen et al., 2015), and their predictions can be manipulated with imperceptible changes in input space (Szegedy et al., 2013). In general, the behavior of deep nets is not well specified when the test queries are out-of-distribution.

A standard practice when studying out-of-distribution detection is to evaluate the detection performance when examples from other datasets are mixed into the test set (Hendrycks & Gimpel, 2016). Here we refer to this type of out-of-distribution input as **out-of-dataset (OOS)**[1] inputs. In the few-shot setting, within each episode, what is in-distribution is specified based on a few labeled examples, known as the *support set*. Hence, there naturally exists another type of out-of-distribution input, the inputs that belong to the same dataset but come from classes not represented by the support

---

[1]We denote **o**ut-**o**f-**d**istribution and **o**ut-**o**f-dataset with the acronyms OOD and OOS, respectively.

set. We refer to these as **out-of-episode (OOE)** examples. These different types of out-of-distribution examples are illustrated in Figure 1.

Being able to detect out-of-distribution examples is critical for improvements in many other important applications, including semi-supervised learning and continual learning. In the case of semi-supervised learning methods, it was shown that if the unlabelled set is polluted with only 25% out-of-distribution examples, then using the unlabeled data actually has a negative effect on performance (Oliver et al., 2018). In the natural continual learning framework, where a model has to learn new concepts while not forgetting old ones, detecting when examples do not belong to any previously-learned class is a fundamental problem.

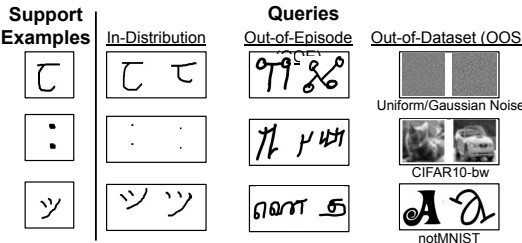

Figure 1: Examples of the support set, in-distribution, **OOE** and **OOS** inputs in one episode.

Hence, in this work, we focus on this core problem of out-of-distribution detection in the few-shot setting.

**Contributions**

- We develop benchmark datasets for out-of-distribution detection, both OOE and OOS, based on four standard benchmark datasets for few-shot classification: Omniglot, CIFAR100, *mini*ImageNet, and *tiered*ImageNet.
- We establish baseline results for both the OOS and OOE tasks for two popular few-shot classifiers—Prototypical Networks and MAML—on these datasets.
- We show that a simple distance metric-based approach dramatically improves the performance on both tasks.
- Finally, we propose a learned scoring function which further improves both tasks on the most challenging new benchmark datasets.

## 2 OVERVIEW OF FEW-SHOT CLASSIFICATION

In few-shot classification, a model is tasked to classify unlabeled 'queries' $Q = \{\mathbf{x}_i\}_{i=1}^{N_Q}$ into one of $N_C$ classes from a set $\mathcal{C}^{test}$. This setup differs from standard 'supervised' classification in that only a few labeled examples are available from each class $c \in \mathcal{C}^{test}$, referred to as that class' *support set* $S_c = \{(\mathbf{x}_i, y_i)\}_{i=1}^{N_S}$. Following the standard terminology, we refer to the number of classes $N_C$ as the 'way' of the task and the number of support examples per class $N_S$ as the 'shot' of the task. We also use the term *episode* to refer to a classification task defined by a support and a query set.

While we assume that little data is available for each such *test* classification episode, the model has access to a (possibly large) training set beforehand that contains examples from a *different set of classes* $\mathcal{C}^{train}$, disjoint from $\mathcal{C}^{test}$. The key is therefore to figure out how to exploit this seemingly-irrelevant data at training time in order to obtain a model that is capable of learning a new episode at test time using only its small support set in a way that performs well on classifying its corresponding query set.

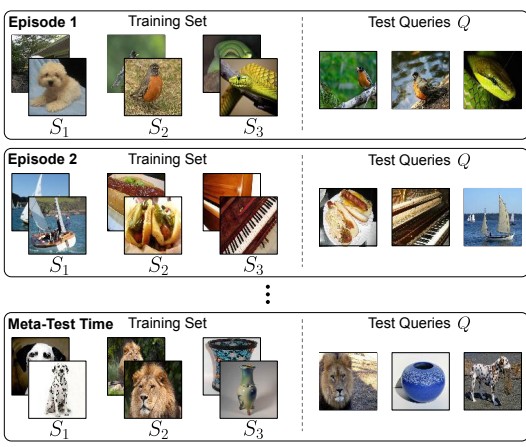

Figure 2: **Standard episodic set-up.** A test *episode* in standard few-shot classification consists of a few training (or support) examples from novel classes, and test/query examples from those classes. Many systems not only evaluate on episodes but also train episodically, i.e., looping over episodes as opposed to over typical mini-batches.

Most recent approaches for this adopt the design choice of creating episodes from the training set of classes too, and expressing the training loss for each episode in terms of performing well on its query examples, after having 'learned' on its small support set. The intuition is to practice learning on episodes that have the same structure as those that will be encountered at test time. At training time, these episodes are created by randomly sampling $N_C$ classes (from the training set of classes), $N_S$ examples of each of those classes to form the support set, and some different examples of each of them to form the query set. We refer to this type of training as 'episodic training' (see Figure 2). Different methods are distinguished by the manner in which learning is performed on the support set. We now give an overview of two popular approaches to few-shot learning: Prototypical Networks (Snell et al., 2017) and MAML (Finn et al., 2017).

**Prototypical Networks.** Prototypical Networks (Snell et al., 2017) are a simple but effective instance of the above framework where the 'learning procedure' that the model undergoes based on the support set has a closed form. More concretely, it consists of a parameterized embedding function $f_\phi$ (typically a deep net) and a distance metric $d(\cdot, \cdot)$ on the embedding space. Given support sets of the chosen classes, the Prototypical Network computes the *prototype* $\boldsymbol{\mu}_c$ of each class $c$ in the embedding space:

$$\boldsymbol{\mu}_c = \frac{1}{|S_c|} \sum_{\mathbf{x}_i \in S_c} f_\phi(\mathbf{x}_i).$$

Then a query $\mathbf{x}^{\text{in}}$ is classified based on its distance to the class prototypes:

$$p_\phi(y = c|\mathbf{x}^{\text{in}}) = \frac{\exp(-d(f_\phi(\mathbf{x}^{\text{in}}), \boldsymbol{\mu}_c))}{\sum_{c'} \exp(-d(f_\phi(\mathbf{x}^{\text{in}}), \boldsymbol{\mu}_{c'}))} \tag{1}$$

During training episodes, the parameters of $f_\phi$ are updated according to the Prototypical Network loss:

$$L_{PN}(\phi; \{S, Q\}) = - \sum_{(\mathbf{x}^{\text{in}}, c) \in Q} \log p_\phi(y = c|\mathbf{x}^{\text{in}}). \tag{2}$$

Algorithm 2 (in Appendix B) is a description of standard *episodic training* of a Prototypical Network.

**Meta-learning.** MAML (Finn et al., 2017) is another popular model of this episodic family that is parameterized by a representation function and a linear classification layer on top, where jointly we denote the weights as $\psi$. Training unfolds over a sequence of training episodes, as usual. In each episode, the weights $\psi$ are adapted via a few steps of gradient descent (denoted as $\text{SGD}_{\text{parameters}}(L)$) to minimize the cross entropy loss over the $N_C$-way classification on the support set, resulting in updated weights $\phi$ which are then used to classify the queries in the given episode. Over a number of episodes, the aggregated loss is then used to update $\psi$ again with gradient descent.

$$\phi_i = \text{SGD}_\psi(\text{CrossEntropyLoss}(\psi; \{S^i\}))$$

$$L_{MAML}(\psi; \{S^i, Q^i\}_{i=1}^M) = \sum_{i=1}^{M} \text{CrossEntropyLoss}(\phi_i; \{S^i, Q^i\})$$

The model is thus encouraged to learn a global initialization $\psi$ of weights such that a few steps of adaptation on a new episode's support set suffice for performing well on its query set.

## 3 OVERVIEW OF OUT-OF-DISTRIBUTION DETECTION

The term "out-of-distribution" refers to input data that is drawn from a different generative process than that of the training data. Hendrycks & Gimpel (2016) used different benchmark datasets as sources of out-of-distribution examples. For example, when a network is trained on MNIST, the out-of-distribution examples can come from Omniglot, black-and-white CIFAR10, etc. Another common evaluation setup is to treat data from the same dataset—but from different classes than those under consideration—as out-of-distribution. These have been referred to as *same manifold* (Liang et al., 2017), or *unobserved class* (Louizos & Welling, 2017) out-of-distribution examples.

**Problem Set-up.** Out-of-distribution detection is a binary detection problem. At test-time, the model is required to produce a score, $s_\theta(\mathbf{x}) \in \mathbb{R}$, where $\mathbf{x}$ is the query, and $\theta$ is the set of learnable parameters for the detection task. We desire $s_\theta(\mathbf{x}^{\text{in}}) > s_\theta(\mathbf{x}^{\text{out}})$, i.e, the scores for in-distribution examples are higher than that of out-of-distribution examples. Typically for quantitative evaluation, threshold-free metrics are used, e.g., the area under the receiver-operating curve (AUROC) (see Section 5 for details).

**Approaches.** The main approaches to out-of-distribution detection can be categorized into one of the three families: 1) scores based on the ***predictive probability*** of a classifier; 2) scores based on fitting a ***density*** model to the inputs directly; and 3) scores based on fitting a density model to ***representations*** of a pretrained model (e.g., a classifier). These are illustrated in Figure 3.

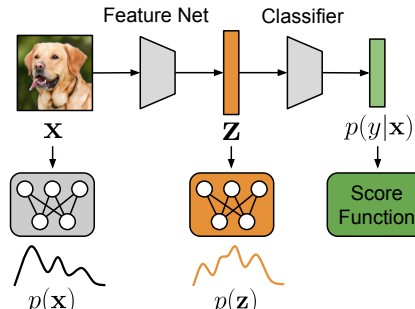

Figure 3: **Schematic of OOS approaches.**

*1. Predictive probability -* Recall that classification of the in-distribution data is done using $p_\phi(y = c|\mathbf{x}^{\text{in}})$ where $\phi$ represents the classifier parameters. Commonly used scores include softmax prediction probability (SPP), $s(\mathbf{x}^{\text{in}}; \phi) = \max_{c'} p_\phi(y = c'|\mathbf{x}^{\text{in}})$ (Hendrycks & Gimpel, 2016) and negative predictive entropy (NPE), $s(\mathbf{x}^{\text{in}}; \phi) = \sum_{c'} p_\phi(y = c'|\mathbf{x}^{\text{in}}) \log p_\phi(y = c'|\mathbf{x}^{\text{in}})$. Note that we use the notation $s(\cdot; \phi)$ to emphasize that these scores operate on top of pretrained classifiers.

A popular extension is to use Bayesian classifiers, i.e., Bayesian Neural Networks (BNNs), and improve the scores by looking at the aggregated score based on the model posterior.

*2. Input density -* Another natural approach to detecting out-of-distribution examples is to fit a density model on the data and consider examples with low likelihood under that model to be OOD. However, this approach is not as competitive when the input domain is high-dimensional images. Nalisnick et al. (2018) showed that deep generative models (e.g., flow-based models (Kingma & Dhariwal, 2018) or auto-regressive models (Salimans et al., 2017)) can assign higher densities to out-of-distribution examples than in-distribution examples.

*3. Representation density -* While fitting a density model on the inputs directly has not proven useful for OOD detection, fitting simple density models on learned classifier representations has. Lee et al. (2018b) fit a Mixture-of-Gaussian (MoG) density with shared diagonal covariance on the classifier activations of the training set. Intuitively, this approach fits a density model in a space where much of the variation in the input has been filtered out, which makes it an easier problem than learning a density model in the input space.

## 4 OUT-OF-DISTRIBUTION DETECTION IN THE FEW-SHOT SETTING

In this study, we focus on two types of out-of-distribution detection problems, described below. In both cases, we denote the set of in-distribution and out-of-distribution examples by $Q = \{\mathbf{x}_i^{\text{in}}\}_{i=1}^{N_Q}$, and $R = \{\mathbf{x}_i^{\text{out}}\}_{i=1}^{N_Q}$ where $N_Q$ is the number of examples. Note that we use $N_Q$ to denote the number of queries in an episode, and the number of in-distribution/out-of-distribution examples, as they mean the same thing depending on context. One could consider different numbers of OOD examples from in-distribution ones, but this is omitted for presentation clarity.

**Out-of-Episode (OOE).** OOE examples come from the same dataset, but from classes not in the current episode. In other words, if the current episode consists of classes in $\mathcal{C}^{episode}$, we sample OOE examples $R$ as follows:

$$\mathcal{C}^{ooe} \leftarrow \text{RANDOMSAMPLE}(\mathcal{C}^{train} \setminus \mathcal{C}^{episode}, N_C) \tag{3}$$

$$R \leftarrow \text{RANDOMSAMPLE}(\mathcal{D}_{\mathcal{C}^{ooe}}, N_Q) \tag{4}$$

Here, $\mathcal{D}_{C'}$ denotes the set of all examples of classes in set $C'$, $\setminus$ is the set difference. This type of out-of-distribution detection is easily motivated. Taking the example where we want to build a

---

**Algorithm 1** Episodic training with OOE inputs. Modified steps are highlighted in blue.

---

1: **while** not converged **do**
2:     $\mathcal{C}^{episode} \leftarrow$ RANDOMSAMPLE$(\mathcal{C}^{train}, N_C)$                         ▷ randomly select classes
3:     **for** $c$ in $\mathcal{C}^{episode}$ **do**                                         ▷ for each class
4:         $S_c \leftarrow$ RANDOMSAMPLE$(\mathcal{D}_{\{c\}}, N_S)$                   ▷ select the support set
5:         $Q_c \leftarrow$ RANDOMSAMPLE$(\mathcal{D}_{\{c\}} \setminus S_c, N_Q)$           ▷ select the query set
6:     **end for**
7:     $\mathcal{C}^{ooe} \leftarrow$ RANDOMSAMPLE$(\mathcal{C}^{train} \setminus \mathcal{C}^{episode}, N_C)$         ▷ prepare OOE queries
8:     $R \leftarrow$ RANDOMSAMPLE$(\mathcal{D}_{\mathcal{C}^{ooe}}, N_Q)$
9:     $\phi \leftarrow \phi - \alpha(\nabla_\phi L_{PN}(\phi; \{S, Q\}) + \lambda \nabla_\phi L_{\text{OOE}}(\phi; \{S, Q, R\})$
10: **end while**

---

customized facial gesture recognizer for a user, when the system sees the user's face performing a gesture that is not registered (i.e., not in the support set), we would like the system to know that the gesture is out-of-distribution, and not perform an inappropriate action.

**Out-of-Dataset (OOS).** OOS examples come from a completely different dataset. For example, if the in-distribution set is Omniglot, then the OOS examples can come from black-and-white CIFAR10. The motivation for this type of out-of-distribution example is also straightforward: a system should defer its actions when faced with something completely different from what it was trained on.

Generally, we use $s(\cdot)$ to denote the scoring function for out-of-distribution detection, which expresses the model's 'confidence' that an example is in-distribution. Hence, we desire that $s(\mathbf{x}^{\text{in}}) > s(\mathbf{x}^{\text{out}})$ for any in-distribution query $\mathbf{x}^{\text{in}}$ and out-of-distribution example $\mathbf{x}^{\text{out}}$.

### 4.1 PROPOSED FEW-SHOT OUT-OF-DISTRIBUTION DETECTION METHODS

In what follows, we propose two novel methods: 1) a parameter-free method that measures the distance in the learned embedding of a few-shot classifier, and 2) a learned scoring function on top of the embedding of a few-shot classifier.

**(1). Minimum Distance Confidence Score (-*MinDist*).** To illustrate why standard softmax prediction probability (SPP) fails in the few-shot setting, consider the classifier learned by Prototypical Network. The original Prototypical Network formulation makes decisions based on a softmax over the negative distances in the embedding space. However, when a query embedding is far away from all prototypes (as we may expect for OOS examples), converting distances to probabilities can yield arbitrarily confident predictions (for details see Appendix E). This makes SPP unsuitable for OOS detection. We propose an alternative confidence score, based on the negative minimum distance from a query to any of the prototypes:

$$s(\mathbf{x}^{\text{in}}; \phi) = - \min_c d(f_\phi(\mathbf{x}^{\text{in}}), \boldsymbol{\mu}_c) \tag{5}$$

**Episodic Optimization with OOE Inputs.** When training our backbone, we can also add a term to our loss to encourage it to accurately detect OOE examples, in addition to accurately performing the episode's classification task. Intuitively, adding this term changes the embedding in such a way that the optimized confidence score performs well on the OOE task. This new term is the following:

$$L_{\text{OOE}}(\phi; \{S, Q, R\}) = - \sum_{\mathbf{x}^{\text{in}} \in Q} \log \sigma(s(\mathbf{x}^{\text{in}}; \phi)) - \sum_{\mathbf{x}^{\text{out}} \in R} \log(1 - \sigma(s(\mathbf{x}^{\text{out}}; \phi))) \tag{6}$$

where $s(\cdot; \phi)$ here can be any of the parameter-free scores, and $\sigma(\cdot)$ is the logistic function. Algorithm 1 is a description of episodic training with OOE examples.

**(2). Learnable Class BOundary (*LCBO*) Network.** We introduce a parametric, class-conditional confidence score that takes a query $\mathbf{x}$ and a class $c$, and yields a score indicating whether $\mathbf{x}$ belongs to class $c$.

The LCBO takes as input: 1) the support embeddings for a particular class, and 2) a query embedding. The LCBO outputs a real-valued score representing the confidence that the query belongs to the corresponding class.

$$\bar{s}_\theta : (\mathbf{x}^{\text{in}}, S_c) \to \mathbb{R} \qquad (7)$$

**Aggregation.** The LCBO outputs class-conditional confidence scores (e.g., the confidence that a query belongs to a specific class). To obtain a final score for in-distribution vs OOS for each query, we aggregate the class-conditional scores. We take the maximum confidence of all the classes:

$$s_\theta(\mathbf{x}^{\text{in}}) = \max_{c \in C} \bar{s}_\theta(\mathbf{x}^{\text{in}}, S_c) \qquad (8)$$

Intuitively, $\bar{s}_\theta(\mathbf{x}^{\text{in}}, S_c)$ computes the distance between a query embedding and a prototype, and the $\max()$ aggregation function says that a query is an inlier if it belongs to at least one class. By design, this is strictly more powerful than -MinDist since it is parameterized by a new set of weights $\theta$, but could also recover simple distance between $\mathbf{x}^{\text{in}}$ and $\boldsymbol{\mu}$, i.e., -MinDist. The difficulty of designing a good uncertainty estimate based on a trained classifier leads us to believe that adding capacity to the confidence score using learnable parameters can be beneficial.

**Implementation Details.** We parameterize the learned confidence score $s_\theta$ by an MLP with two hidden layers of dimension 100, that takes as input the concatenation $[\boldsymbol{\mu}_c; f_\phi(\mathbf{x}^{\text{in}})]$ where $\boldsymbol{\mu}_c$ is the class prototype and $f_\phi(\mathbf{x}^{\text{in}})$ is the query embedding. Note that LCBO always operates on top of the backbone $f_\phi(\cdot)$, so this dependency is omitted for notational simplicity.

**Training the LCBO.** We train the LCBO episodically. However, instead of training the aggregated score, we use the following binary cross-entropy objective on the score before aggregation:

$$L_{\text{LCBO}}(\phi, \theta; \{S, Q, R\}) = - \sum_{(c, \mathbf{x}^{\text{in}}) \in Q} \log \sigma(\bar{s}_\theta(\mathbf{x}^{\text{in}}, S_c)) - \sum_{\mathbf{x}^{\text{out}} \in R, c' \sim \text{unif}(\mathcal{C}^{episode})} \log(1 - \sigma(\bar{s}_\theta(\mathbf{x}^{\text{out}}, S_{c'}))) \qquad (9)$$

For the OOE queries $\mathbf{x}^{\text{out}}$, we assigned them a label drawn from the uniform distribution of the in-distribution classes.

## 5 Experiments

In this section, we: 1) establish the OOE and OOS detection performance of standard few-shot methods as well as a novel variant, and 2) show that both our proposed methods improve substantially over these baseline approaches.

To enable fair comparisons, for the experiments in this section we use the same network configuration, a standard 4-layer ConvNet architecture that is well-established in the few-shot literature (Snell et al., 2017). None of the methods discussed here sacrifice in-distribution classification accuracy.

**Evaluation Metrics.** We evaluate the OOE and OOS detection performance using the area under the receiver-operating curve (AUROC). This is a simple metric that circumvents the need to set a threshold for the score. The base-rate (i.e., a completely naïve scoring function) for all of our experiments is 50%. A scoring function that can completely separate $s(\mathbf{x}^{\text{in}})$ from $s(\mathbf{x}^{\text{out}})$ would achieve an AUROC score of 100%. Following standard practice (Hendrycks & Gimpel, 2016; Liang et al., 2017; Lee et al., 2018a), we also report scores for area under the precision and recall curve (AUPR), and false positive rate (FPR) (Table 1). All results are evaluated using 1000 *test episodes*, i.e., episodes that contain classes never seen during training. Please refer to Appendix C for descriptions of the in-distribution and OOS datasets.

### 5.1 Out-of-Distribution Detection with Baseline methods

We first evaluate the out-of-episode and out-of-distribution detection performance of three few-shot classifiers, using the standard SPP confidence score. The results are summarized in Table 1. We note that not only are these classifiers similar in their distribution classification accuracy (Chen et al., 2019), but their ability to detect out-of-distribution examples is also similar.

| in-distribution | task | AUROC ↑ | | | AUPR ↑ | | | FPR90 ↓ | | |
|---|---|---|---|---|---|---|---|---|---|---|
| | | PN | MAML | ABML | PN | MAML | ABML | PN | MAML | ABML |
| Omniglot | OOE | 90.6 | 88.4 | 85.5 | 90.3 | 89.3 | 86.3 | 27.5 | 38.1 | 46.1 |
| | OOS | 63.8 | 89.5 | 88.8 | 70.8 | 90.3 | 89.1 | 53.7 | 35.1 | 38.3 |
| CIFAR100 | OOE | 60.3 | 61.6 | 59.4 | 63.0 | 63.6 | 61.1 | 85.1 | 84.0 | 85.3 |
| | OOS | 57.4 | 58.1 | 65.8 | 63.8 | 62.6 | 68.1 | 86.7 | 86.0 | 79.3 |
| *mini*ImageNet | OOE | 56.6 | 56.8 | 54.6 | 58.5 | 58.1 | 56.0 | 87.0 | 86.9 | 88.4 |
| | OOS | 50.4 | 65.8 | 63.8 | 61.5 | 68.2 | 65.4 | 0.4 | 78.5 | 80.0 |
| *tiered*ImageNet | OOE | 59.4 | 57.3 | 63.3 | 61.3 | 58.6 | 66.3 | 85.1 | 86.5 | 80.7 |
| | OOS | 66.4 | 68.3 | 64.2 | 72.6 | 70.3 | 65.4 | 79.0 | 75.7 | 81.4 |

Table 1: **Baseline results for various few-shot classifiers.** All numbers are in percentages, evaluated in the 5-way 5-shot setting for all 4 datasets, using the standard Conv4 backbone and SPP confidence score. The reported OOS numbers are the means over all the OOS datasets used for the corresponding in-distribution dataset.

| in-distribution | task | AUROC ↑ | | | AUPR ↑ | | | FPR90 ↓ | | |
|---|---|---|---|---|---|---|---|---|---|---|
| | | SPP | -MinDist | LCBO | SPP | -MinDist | LCBO | SPP | -MinDist | LCBO |
| Omniglot | OOE | 89.5 | **98.3** | 96.4 | 88.6 | **98.2** | 92.5 | 28.3 | **3.8** | 7.3 |
| | OOS | 35.8 | **100** | 58.4 | 45.6 | **100** | 58.5 | 80.4 | **0.0** | 42.1 |
| CIFAR100 | OOE | 60.1 | 68.0 | **73.3** | 61.0 | 67.2 | **71.5** | 84.3 | 73.1 | **62.8** |
| | OOS | 55.8 | **86.1** | 79.6 | 58.3 | **86.2** | 79.3 | 87.6 | **31.9** | 52.5 |
| *mini*ImageNet | OOE | 56.7 | 61.9 | **65.6** | 56.8 | 61.1 | **63.1** | 86.8 | 80.2 | **73.1** |
| | OOS | 51.8 | 61.0 | **74.7** | 54.0 | 64.0 | **75.2** | 89.2 | **60.1** | 61.0 |
| *tiered*ImageNet | OOE | 59.0 | 62.4 | **65.0** | 60.0 | 61.4 | **62.8** | 85.1 | 79.0 | **74.4** |
| | OOS | 53.6 | 51.4 | **70.7** | 56.2 | 59.9 | **73.1** | 88.5 | **66.0** | 70.4 |

Table 2: **LCBO, -MinDist results for ProtoNet.** All numbers are in percentages, evaluated in the 5-way 5-shot setting for all 4 datasets, using the standard Conv4 backbone. The reported OOS numbers are the means over all the OOS datasets used for the corresponding in-distribution dataset. For detailed OOS results, see Appendix D

Bayesian methods provide an alternative that may help in OOD detection, by quantifying uncertainty in predictions. We evaluate a recent method that shows strong calibration results: the Amortized Bayesian Meta-Learning (ABML) algorithm (Ravi & Beatson, 2019) which realizes a Bayesian MAML following the hierarchical variational Bayes formulation of Amit & Meir (2018). However, ABML did not significantly improve over MAML, at least according to our implementation (since Ravi & Beatson (2019) did not release code, in App. F we discuss details of our best effort to reproduce this method). Next we show that out-of-distribution performance can be greatly improved.

## 5.2 OUT-OF-DISTRIBUTION DETECTION WITH -MINDIST & LCBO

Few-shot classification can be evaluated in many different (*way*, *shot*) settings, e.g., 5-way 5-shot, 10-way 1-shot, etc. Due to lack of space, we report only 5-shot 5-way results in this section. Full results for $\{5, 10\}$-way $\times \{1, 5\}$-shot settings on CIFAR-100 are provided in Appendix I.

Table 2 shows the results of SPP, -MinDist, and the learned LCBO score on all four of the datasets. Across the board, on both OOE and OOS tasks, either -MinDist or LCBO outperformed the baseline method. Interestingly, it seems to confirm our hypothesis that -MinDist might not be the most suitable confidence score for all embedding spaces. For a more detailed discussion of -MinDist and its connection to a similar method proposed in the supervised setting (Lee et al., 2018b), please see Appendix E. On the largest datasets, i.e., both versions of the ImageNet dataset, LCBO outperformed -MinDist on both OOE and OOS tasks. This was somewhat surprising, since one might expect that parameter-free functions like -MinDist can generalize better to OOS datasets that are very different from the in-distribution data. This was still true on CIFAR100, but not on ImageNet datasets.

One major difference between CIFAR100 and ImageNet was the image size ($32 \times 32$ vs $84 \times 84$), which resulted in different embedding dimensions (256 vs 1600). This suggests that as we scale up the dimensionality of the embedding space, it becomes increasingly difficult to design a suitable parameter-free confidence score. Hence, a learnable score such as LCBO becomes critical.

**Effect of different backbones.** We also investigated the effect of the backbone network, $f_\phi$, on OOE and OOS detection. Recently, Chen et al. (2019) trained larger backbones like ResNet without using episodic training. We include results with these larger backbones in Appendix I.

**Effect on example downstream application.**

Ren et al. (2018) proposed to study few-shot semi-supervised learning (FS-SSL), where each episode is augmented with an *unlabelled set*. To make it more realistic, there are also 'distractors' present.

| Model | OOE | Uniform | Gaussian |
|---|---|---|---|
| Supervised | 47.5 | 47.7 | 47.4 |
| Soft $k$-means (Ren et al., 2018) | +1.4 | −9.4 | −11.4 |
| with LCBO | −0.2 | +0.1 | +0.0 |

Table 3: Classification accuracy (in percentages) of semi-supervised learning results on *tiered*ImageNet. Column headings indicate type of distractor used at test-time. '+', and '−' denote the lack and presence of degradation. 'Supervised' refers to training without an unlabelled set.

In previous FS-SSL studies, only OOE examples are considered for both training and testing phases. This is somewhat unrealistic, as there can be unforeseen distractors in the test episodes. In Table 3 we show that when evaluated in this more realistic setting, the method of Ren et al. (2018) suffers. Here, we do not claim that LCBO improves upon semi-supervised learning methods. Still, especially in the case when distractor inputs are OOS instead of only OOE examples, baseline semi-supervised methods significantly degrade the classification accuracy (see Appendix G for more details on this task).

## 6    RELATED WORK

As few-shot out-of-distribution detection is a new problem, here we discuss recent attempts to study uncertainty in the few-shot setting, and previous approaches that worked well for out-of-distribution detection in the supervised setting.

**Bayesian Few-Shot Classifier.** A number of papers investigated Bayesian extensions of MAML. Compared with other works (Grant et al., 2018; Finn et al., 2018; Yoon et al., 2018) on extending the MAML framework to the Bayesian setting, ABML maintains uncertainty on the global initialization $\psi$. Furthermore, as a way to analyze the uncertainty estimation of ABML, Ravi & Beatson (2019) studied the few-shot out-of-distribution detection of the ABML framework. However, they did not report quantitative evaluations as we did.

**Other Out-of-Distribution Approaches.** ODIN (Liang et al., 2017) consists of 2 innovations: 1) it performs temperature scaling to calibrate the predicted probability (Guo et al., 2017); and 2) when doing out-of-distribution detection, it adds virtual adversarial perturbations (VAP) to the input. Intuitively, VAP will have a larger effect on the in-distribution input compared to the out-of-distribution input. Lee et al. (2018b) showed that this approach can be complementary to fitting a Gaussian density to the activations of the network. Our preliminary experiments showed that ODIN did not have a big impact in the few-shot setting. Outlier exposure, another recent method (Hendrycks et al., 2019), also did not show a significant effect. We included these results in Appendix J.

For a while, methods using the ***predictive probability*** were the dominant approach in out-of-distribution detection. Nalisnick et al. (2018) pointed out that the community had been using the learned density model incorrectly by directly looking at the $p(\mathbf{x})$ scores, and instead should use a measure of typicality (Nalisnick et al., 2019). Ren et al. (2019) proposed to train a separate "background" model and use the likelihood ratio as the score. Generative/density models have not been extensively studied in the few-shot setting. We believe that this is due to the lack of a good task/quantitative evaluation, and that the tasks we study might facilitate research done on such models. Another topic similar to ours is generalized zero-shot recognition (Mandal et al., 2019). The main difference in our setting is that not only are the OOD examples unseen, but the in-distribution examples/classes at evaluation time are also unseen, and only defined by a support set.

## 7    CONCLUSION

To the best of our knowledge, this is the first study to investigate both OOS and OOE tasks and report results using commonly-used metrics in the few-shot setting. We showed that existing confidence scores developed in the supervised setting (i.e., setting with a fixed number of classes) are not suitable when used with popular few-shot classifiers. Our proposed confidence scores, -MinDist and LCBO, substantially outperformed the baselines on both tasks across four staple few-shot classification datasets. We hope that our work encourages future studies on quantitative evaluation of out-of-distribution detection and uncertainty in the few-shot setting.

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

## A    NOTATION

| Symbol | Meaning |
|:---:|:---:|
| $Q, S, R$ | query, support, distractors/out-of-distribution sets |
| $\mathcal{C}^{test}, \mathcal{C}^{train}$ | classes in the test, train set |
| $\mathcal{C}^{episode}$ | classes in an episode |
| $\mathcal{D}_{\mathcal{C}^{episode}}$ | the set of all examples belonging to class $\mathcal{C}^{episode}$ |
| $N_C, N_S$ | number of way/classes, number of shots per episode |
| $\mathbf{x}$ | generic image input |
| $\mathbf{x}^{in}, \mathbf{x}^{out}$ | query, and out-of-distribution examples |
| $S_c$ | the set of support examples of class $c$ |
| $f_\phi$ | embedding/backbone network |
| $\boldsymbol{\mu}$ | prototype |
| $s(\cdot)$ | confidence score |

Table 4: Description of the functions used throughout this paper.

## B    EPISODE CONSTRUCTIONS

**Training.**    Algorithm 2 is a description of episodic training of a classifier. Here, $\mathcal{D}_{C'}$ denotes the set of all examples of classes in set $C'$. RANDOMSAMPLE$(s, n)$ randomly selects $n$ elements from the set $s$.

---
**Algorithm 2** Episodic training.

---
1: **while** not converged **do**
2:     $\mathcal{C}^{episode} \leftarrow$ RANDOMSAMPLE$(\mathcal{C}^{train}, N_C)$      ▷ sample classes
3:     **for** $c$ in $\mathcal{C}^{episode}$ **do**      ▷ for each class
4:         $S_c \leftarrow$ RANDOMSAMPLE$(\mathcal{D}_{\{c\}}, N_S)$      ▷ sample support set
5:         $Q_c \leftarrow$ RANDOMSAMPLE$(\mathcal{D}_{\{c\}} \setminus S_c, N_Q)$      ▷ sample query set
6:     **end for**
7:     $\phi \leftarrow \phi - \alpha(\nabla_\phi L_{PN}(\phi; \{S, Q\}))$
8: **end while**

---

**Evaluation.**    Algorithm 3 is a description of episodic evaluation of the OOE task. Note that now both the in-distribution and out-of-episode classes are drawn from unseen classes. Both the in-distribution and out-of-episodes scores are accumulated over 1000 episodes, and evaluated using a metric such as AUROC. In the OOS setting, one would modify lines 9 to 12 to sample from the OOS set, e.g., $R \leftarrow$ RANDOMSAMPLE$(\mathcal{D}_{OOS}, N_C * N_Q)$.

## C    DATASETS

**Omniglot.**    The Omniglot dataset (Lake et al., 2011) contains $28 \times 28$ greyscale images of handwritten characters. This is the most widely adopted benchmark dataset for few-shot classification. We use the same splits as in (Snell et al., 2017). Each class has 20 samples, and there are a total of $1200 \times 4$ training classes and $423 \times 4$ unseen classes.

**CIFAR100.**    The CIFAR100 dataset (Krizhevsky, 2009) contains $32 \times 32$ color images. It is similar to the CIFAR10 dataset, but has 100 classes of 600 images each. We used 64 classes for training, 16 for validation, and 20 for test.

**_mini_ImageNet.**    The _mini_ImageNet dataset is another commonly used few-shot benchmark (Snell et al., 2017; Vinyals et al., 2016). It consists of $84 \times 84$ colored images. It also has 100 classes, and 600 examples each. Similarly, we used 64 classes for training, 16 for validation, and 20 for test.

---

**Algorithm 3** Episodic Evaluation (OOE) .

---

1: $S_{in} \leftarrow \varnothing$
2: $S_{out} \leftarrow \varnothing$
3: **for** 1000 **do**
4:     $\mathcal{C}^{episode} \leftarrow \text{RANDOMSAMPLE}(\mathcal{C}^{test}, N_C)$             ▷ sample in-distribution examples
5:     **for** $c$ in $\mathcal{C}^{episode}$ **do**
6:         $S_c \leftarrow \text{RANDOMSAMPLE}(\mathcal{D}_{\{c\}}, N_S)$
7:         $Q_c \leftarrow \text{RANDOMSAMPLE}(\mathcal{D}_{\{c\}} \setminus S_c, N_Q)$
8:     **end for**
9:     $\mathcal{C}^{ooe} \leftarrow \text{RANDOMSAMPLE}(\mathcal{C}^{test} \setminus \mathcal{C}^{episode}, N_C)$         ▷ sample OOE examples
10:    **for** $c$ in $\mathcal{C}^{ooe}$ **do**
11:        $R_c \leftarrow \text{RANDOMSAMPLE}(\mathcal{D}_{\{c\}}, N_Q)$
12:    **end for**
13:    **for** $\mathbf{x}^{in}$ in $Q$ **do**
14:       $S_{in} \leftarrow S_{in} \cup \{s(\mathbf{x}^{in}; \phi)\}$
15:    **end for**
16:    **for** $\mathbf{x}^{out}$ in $R$ **do**
17:       $S_{out} \leftarrow S_{out} \cup \{s(\mathbf{x}^{out}; \phi)\}$
18:    **end for**
19: **end for**
20: $\text{Metric}(S_{in}, S_{out})$

---

***tiered*Imagenet.** The *tiered*ImageNet dataset is very similar to the *mini*ImageNet dataset. Proposed by Ren et al. (2018), it has 608 classes instead of 100. [2]

**Out-of-Dataset.** The OOS datasets were adopted from previous studies including those by Hendrycks et al. (2019); Liang et al. (2017). Since we experimented with in-distribution datasets of different scales, the OOS inputs were scaled accordingly.

- **Noise:** We used uniform, Gaussian, and Rademacher noise of the same dimensionality as the in-distribution data (e.g., $3 \times 32 \times 32$ uniform noise as OOS data for CIFAR-100).

- **notMNIST** consists of $28 \times 28$ grayscale images of alphabetic characters from several typefaces.

- **CIFAR10bw** is simply a grayscale version of CIFAR10.

- **LSUN** is a large-scale scene understanding dataset (Yu et al., 2015).

- **iSUN** is a subset of SUN consisting of 8925 images (Xu et al., 2015).

- **Texture** is a dataset with different real world patterns (Cimpoi et al., 2014).

- **Places** is another large scale scene understanding dataset (Zhou et al., 2017).

- **SVHN** refers to the Google Street View House Numbers dataset (Netzer et al., 2011).

- **TinyImagenet** consists of $64 \times 64$ color images from 200 ImageNet classes, with 600 examples of each class.

## D  EXPANDED TABLE 2

All the results in this section are in the 5-way, 5-shot setting, and were obtained using the 4-layer convolutional backbone.

---

[2]We follow the instructions on https://github.com/renmengye/few-shot-ssl-public.

## D.1 OMNIGLOT

| Metric | AUROC↑ | | | AUPR↑ | | | FPR90↓ | | |
|---|---|---|---|---|---|---|---|---|---|
| Method | SPP | -MinDist | LCBO | SPP | -MinDist | LCBO | SPP | -MinDist | LCBO |
| OOE | 89.5 | **98.2** | 96.4 | 88.6 | **98.3** | 92.5 | 28.3 | **3.8** | 7.3 |
| Gaussian | 17.4 | **100.0** | 82.9 | 34.3 | **100.0** | 67.2 | 95.5 | **0.0** | 17.7 |
| uniform | 86.5 | **100.0** | 98.2 | 84.6 | **100.0** | 98.5 | 36.2 | **0.0** | 2.5 |
| notMNIST | 28.4 | **100.0** | 12.2 | 37.4 | **100.0** | 33.2 | 87.7 | **0.0** | 88.5 |
| cifar10bw | 29.5 | **100.0** | 28.7 | 37.7 | **100.0** | 37.5 | 86.9 | **0.0** | 71.7 |
| MNIST | 17.1 | **100.0** | 70.1 | 34.2 | **100.0** | 56.1 | 95.6 | **0.0** | 30.1 |
| OOS MEAN | 35.8 | **100.0** | 58.4 | 45.6 | **100.0** | 58.5 | 80.4 | **0.0** | 42.1 |
| MEAN | 44.7 | **99.7** | 64.8 | 52.8 | **99.7** | 64.2 | 71.7 | **0.6** | 36.3 |

Table 5: Expanded Omniglot results

## D.2 CIFAR100

| Metric | AUROC↑ | | | AUPR↑ | | | FPR90↓ | | |
|---|---|---|---|---|---|---|---|---|---|
| Method | SPP | -MinDist | LCBO | SPP | -MinDist | LCBO | SPP | -MinDist | LCBO |
| OOE | 60.1 | 68.0 | **73.3** | 61.0 | 67.2 | **71.5** | 84.3 | 73.1 | **62.8** |
| Gaussian | 47.2 | **100.0** | 89.0 | 49.7 | **100.0** | 88.5 | 93.3 | **0.0** | 30.9 |
| Uniform | 63.7 | 96.2 | 82.8 | 67.6 | 97.1 | 83.9 | 83.8 | 7.5 | 48.8 |
| Rademacher | 47.3 | **100.0** | 87.2 | 49.6 | **100.0** | 86.5 | 93.0 | **0.0** | 34.7 |
| Texture | 54.5 | 89.5 | 80.4 | 54.8 | 87.5 | 78.6 | 87.9 | 26.1 | 48.9 |
| Places | 56.5 | 88.9 | 78.1 | 58.4 | 89.0 | 77.8 | 87.9 | 32.9 | 56.8 |
| SVHN | 64.0 | 48.4 | 67.7 | 67.0 | 50.7 | 68.2 | 82.2 | 93.1 | 75.5 |
| LSUN | 57.3 | 91.4 | 79.5 | 59.2 | 91.7 | 80.3 | 87.5 | 25.7 | 56.7 |
| iSUN | 55.6 | 90.1 | 78.7 | 57.4 | 90.1 | 78.6 | 88.4 | 28.9 | 56.4 |
| TinyImagenet | 56.2 | 88.9 | 79.7 | 58.0 | 88.5 | 79.5 | 88.1 | 31.4 | 53.7 |
| OOS MEAN | 55.8 | **88.2** | 80.3 | 58.0 | **88.3** | 80.2 | 88.0 | **27.3** | 51.4 |
| MEAN | 56.2 | **86.1** | 79.6 | 58.3 | **86.2** | 79.3 | 87.6 | **31.9** | 52.5 |

Table 6: Expanded CIFAR100 results

## D.3 *mini*IMAGENET

| Metric | AUROC↑ | | | AUPR↑ | | | FPR90↓ | | |
|---|---|---|---|---|---|---|---|---|---|
| Method | SPP | -MinDist | LCBO | SPP | -MinDist | LCBO | SPP | -MinDist | LCBO |
| OOE | 56.7 | 61.9 | **65.6** | 56.8 | 61.1 | **63.1** | 86.8 | 80.2 | **73.1** |
| Gaussian | 37.4 | **100.0** | 64.3 | 41.7 | **100.0** | 64.7 | 95.8 | **0.0** | 68.0 |
| Uniform | 54.4 | 99.8 | 87.8 | 56.3 | 99.8 | 87.3 | 87.5 | **0.0** | 34.4 |
| Rademacher | 39.0 | **100.0** | 64.0 | 42.4 | **100.0** | 65.0 | 95.7 | **0.0** | 71.1 |
| Texture | 52.7 | 49.9 | 74.6 | 53.7 | 45.9 | 73.3 | 88.8 | 77.5 | 60.2 |
| Places | 57.7 | 46.6 | 76.6 | 59.0 | 47.7 | 77.3 | 86.1 | 91.3 | 61.6 |
| SVHN | 51.1 | 5.6 | 74.5 | 54.0 | 31.2 | 76.2 | 91.0 | **100.0** | 65.8 |
| LSUN | 59.2 | 51.3 | 76.1 | 61.4 | 53.6 | 78.2 | 85.2 | 92.7 | 66.4 |
| iSUN | 57.9 | 49.7 | 78.1 | 59.4 | 50.2 | 78.7 | 85.6 | 89.5 | 59.4 |
| TinyImagenet | 56.4 | 46.5 | 75.9 | 57.7 | 47.2 | 76.0 | 86.9 | 90.1 | 62.0 |
| OOS MEAN | 51.8 | 61.0 | **74.7** | 54.0 | 64.0 | **75.2** | 89.2 | **60.1** | 61.0 |
| MEAN | 52.2 | 61.1 | **73.8** | 54.2 | 63.7 | **74.0** | 88.9 | **62.1** | 62.2 |

Table 7: Expanded *mini*ImageNet results.

## D.4 *tiered*IMAGENET

| Metric | AUROC↑ | | | AUPR↑ | | | FPR90↓ | | |
|---|---|---|---|---|---|---|---|---|---|
| Method | SPP | -MinDist | LCBO | SPP | -MinDist | LCBO | SPP | -MinDist | LCBO |
| OOE | 59.0 | 62.4 | **65.0** | 60.0 | 61.4 | **62.8** | 85.1 | 79.0 | **74.4** |
| Gaussian | 38.4 | **100.0** | 76.2 | 41.5 | **100.0** | 77.7 | 95.6 | **0.0** | 57.6 |
| Uniform | 41.6 | 99.1 | 90.4 | 42.8 | 98.9 | 92.0 | 94.0 | 2.0 | 32.5 |
| Rademacher | 40.0 | **100.0** | 77.3 | 42.3 | **100.0** | 78.1 | 95.1 | **0.0** | 54.2 |
| Texture | 55.5 | 34.5 | 61.9 | 56.8 | 40.6 | 63.5 | 88.6 | 93.3 | 81.8 |
| Places | 61.7 | 27.9 | 66.9 | 64.7 | 40.6 | 70.0 | 84.6 | 99.7 | 80.4 |
| SVHN | 54.9 | 10.6 | 57.8 | 58.9 | 32.2 | 61.5 | 90.5 | 99.9 | 90.0 |
| LSUN | 66.8 | 30.6 | 71.4 | 70.0 | 43.8 | 75.0 | 79.9 | 99.8 | 77.2 |
| iSUN | 62.7 | 29.4 | 67.9 | 65.1 | 41.4 | 71.0 | 82.8 | 99.6 | 79.9 |
| TinyImagenet | 60.7 | 30.8 | 66.5 | 63.3 | 42.0 | 69.2 | 85.2 | 99.4 | 80.0 |
| OOS MEAN | 53.6 | 51.4 | **70.7** | 56.2 | 59.9 | **73.1** | 88.5 | **66.0** | 70.4 |
| MEAN | 54.1 | 52.5 | **70.1** | 56.5 | 60.1 | **72.1** | 88.1 | **67.3** | 70.8 |

Table 8: Expanded *tiered*ImageNet results.

## E  -MINDIST

In Tables 5 and  6, we show that -MinDist improved both OOE and OOS detection results under all metrics. The improvement on the OOS task was very pronounced due to the fact that baseline scoring functions based on $p(y|\mathbf{x}^{\text{in}})$ behaved erratically for $\mathbf{x}^{\text{in}}$ far away from the empirical distribution of the in-distribution embedding. For example, when the embedding network is trained on CIFAR100, an embedded point based on image of Gaussian noise has an $L_2$-norm $10\times$ larger than the average embedding of an in-distribution input. This resulted in a very confident SPP score (see paragraph below). This effect was eliminated by using -MinDist, and any embedded point far away from the class prototypes was assigned low confidence. This intuition seemed to apply to most of the OOS tasks. On the more challenging task of OOE detection, -MinDist improved over the baselines, but not by as large a margin when the in-distribution dataset was easy (e.g., Omniglot). The improvement on the OOE task was more substantial when the in-distribution dataset was CIFAR100.

**Toy example of when 'softmax of distance' breaks down.**  Note that when the input to the softmax, or our *logits*, are the negative distances to each of the prototypes:

$$p_\phi(y = c|\mathbf{x}^{\text{in}}) = \frac{\exp(-d(f_\phi(\mathbf{x}^{\text{in}}), \boldsymbol{\mu}_c))}{\sum_{c'} \exp(-d(f_\phi(\mathbf{x}^{\text{in}}), \boldsymbol{\mu}_{c'}))} \tag{10}$$

the softmax function is invariant to a constant additive bias in the logits. This makes anything outside of the convex hull formed by the prototypes equivalent to being right on the boundary of the convex hull. In the case that we have a 1-dimensional embedding, and only 2 prototypes located at 0 and 1, anything within the range of 0 and 1 would give reasonable probability, and the point 0.5 would give maximum entropy. However, our intuition says that anything that is very far away from both prototypes, say the point of 100, should also have maximum entropy. Yet, due to the invariant to constant additive bias, anything outside of the range 0 and 1 would have the undesirable behavior that as one moves away from this range, the output of the softmax decreases in entropy while we desire it to increase in entropy. In higher dimensions, a similar phenomenon happens, hence confidence functions that operate in the predicted probability space are not suitable for the out-of-distribution data.

A good connection between -MinDist and the method in (Lee et al., 2018b) can be made. However, Lee et al. (2018b) fit a full covariance Gaussian to each of the classes, and use the Mahalanobis distance as score, which requires computing the inverse covariance of the support embeddings. This approach faces a fundamental difficulty in the few-shot setting: because the number of training examples (i.e., 25 for the 5-way 5-shot setting)

Table 9: AUROC comparison to (Lee et al., 2018b)

| Dataset | CIFAR100 | | *mini*ImageNet | |
|---|---|---|---|---|
| Model | OOE | OOS | OOE | OOS |
| **-MinDist** | 68 | **86** | 62 | 61 |
| **LCBO** | **73** | 80 | **66** | **75** |
| Mahalanobis (tied) (Lee et al., 2018b) | 57 | 86 | 53 | 59 |
| Mahalanobis (Lee et al., 2018b) | 54 | 86 | 56 | 42 |

```
In [9]: a = .5

In [10]: F.softmax(torch.FloatTensor([a-0,a-1]).pow(2), 0)
Out[10]: tensor([0.5000, 0.5000])

In [11]: a = 100

In [12]: F.softmax(torch.FloatTensor([a-0,a-1]).pow(2), 0)
Out[12]: tensor([1., 0.])
```

Figure 4: The toy example in PyTorch. $a$ is our embedded query, and we have a prototype at 0, and another at 1. When $a = .5$, SPP is 0.5. When $a = 100$, SPP is 1, which is undesirable.

is smaller than the dimension of the embedding space (i.e. 256), the covariance matrix is singular. Early in our project we found that the most natural adaptation of (Lee et al., 2018b), which learns a Gaussian with diagonal covariance per class, performed worse than -MinDist (Table 9).

## F  ABML

The setup of few-shot ABML consists of a prior $p(\psi)$ on the global initialization $\psi$, and a prior $p(\phi|\psi)$ on episode-specific model weights for each episode. The training objective is to learn a posterior distribution of $\psi$ which maximizes a variational lower bound of the likelihood of the data.

In each episode, with model weights prior $p(\phi|\psi)$, the ABML algorithm performs standard Bayes by Backprop (Blundell et al., 2015) on the support set to obtain the variational posterior distribution for $\phi$. In practice, the initial variational parameter for $\phi$ is set to $\psi$ to reduce the total number of parameters, while the performance did not seem to be negatively affected empirically (Ravi & Beatson, 2019). Furthermore, based on the assumption that the variance in $\psi$ should be low due to training over a large number of episodes, Ravi & Beatson (2019) simplify the inference of $\psi$ to a point estimate, and update $\psi$ by the usual gradient descent with gradients aggregated over a sequence of episodes, analogous to the MAML setting.

Following the description in (Ravi & Beatson, 2019), we implemented ABML based on the MAML implementation we got from https://github.com/wyharveychen/CloserLookFewShot.

|  | Suggested in | |
|---|---|---|
|  | Ravi & Beatson (2019) | Ours |
| Inner LR | 0.1 | 0.01 |
| Outer LR | 0.001 | 0.001 |
| SGD steps (training) | 5 | 5 |
| SGD steps (testing) | 10 | 10 |
| Num posterior samples (train-inner) | 5 | 1 |
| Num posterior samples (train-outer) | 2 | 1 |
| Num posterior samples (test) | 10 | 10 |
| $a_0$ for hyper-prior | 2 | 2 |
| $b_0$ for hyper-prior | .2 | .2 |
| Inner KL weight | ? | 0.01 |
| Outer KL weight | ? | 0.1 |

Table 10: Hyperparameters used for ABML. The last two rows, the KL weights, are not described in (Ravi & Beatson, 2019) explicitly, but only described as 'down-weighed' in their text. We chose what empirically works best for us.

Since in general it is difficult to measure how properly Bayesian a method is, we also performed the calibration experiment from the original paper, and found a similar trend (see Figure 5). Combined with similar classification accuracy, we believe that we have a somewhat meaningful implementation of ABML.

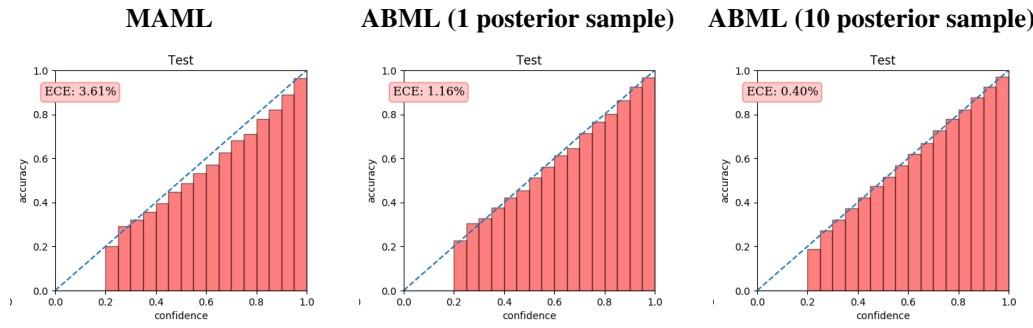

Figure 5: Calibration results. ABML with 10 posterior samples (ECE=0.40%) have better calibration than ABML with 1 posterior sample (ECE=1.16%), and MAML (ECE=3.61%). ECE is the expected calibration error (Guo et al., 2017).

## G    SEMI-SUPERVISED FEW-SHOT CLASSIFICATION

First studied by Ren et al. (2018), there has been a recent surge of interest in semi-supervised few-shot learning. Each episode has an additional unlabeled set $U = \{\mathbf{u}\}_i^{N_u}$. Examples from this set act as additional learning signals in each episode, much like the role of the support set. However, there are two differences: 1) we are not given label information, and 2) it contains 'distractor' classes, i.e., data that do not come from target classes of interest. In (Ren et al., 2018), their 'distractor' inputs are exactly what we refer to as OOE inputs here.

It is known, at least in the supervised setting, that when the unlabelled dataset is polluted with out-of-distribution examples, semi-supervised methods can sometimes even degrade the classifier accuracy (Oliver et al., 2018). Similarly, in (Ren et al., 2018), without the more sophisticated methods that implicitly mask out distractors, soft $k$-Means with the unlabelled dataset barely has an effect.

Here, we propose a simple semi-supervised inference method with Prototypical Networks based on LCBO. Since naturally, we can think of $p_{i,c} \triangleq \sigma(\bar{s}_\theta(\mathbf{u}_i, S_c))$ as the probability of an unknown input $\mathbf{u}_i$ belonging to class $c$, we simply perform soft $k$-Means to obtain our new prototypes using $\tilde{p}_{i,c}$ as the responsibilities:

$$\tilde{\boldsymbol{\mu}}_c = \frac{\sum_{\mathbf{x}_i \in S_c} f_\phi(\mathbf{x}) + \sum_{\mathbf{u}_i \in U} \tilde{p}_i \tilde{p}_{i,c} f_\phi(\mathbf{u})}{|S_c| + \sum_{\mathbf{u}_i \in U} \tilde{p}_i \tilde{p}_{i,c}} \tag{11}$$

$$\tilde{p}_i = \frac{\max_c \tilde{p}_{i,c}}{\sum_i \max_c \tilde{p}_{i,c}} \tag{12}$$

$$\tilde{p}_{i,c} = ReLU(p_{i,c} - .5) \tag{13}$$

and classification in this semi-supervised setting is done based on these updated prototypes $\tilde{\boldsymbol{\mu}}_c$. We use $\tilde{p}_{i,c}$ instead of $p_{i,c}$ because $p_{i,c}$ was optimized so that a point on the boundary of being out-of-distribution would have a $p_{i,c}$ of .5, whereas in this soft clustering scheme, we want those points to have 0 weight.

| Model | OOE | Uniform | Gaussian |
|---|---|---|---|
| Supervised | 47.5 | 47.7 | 47.4 |
| Baseline soft $k$-means (Ren et al., 2018) | **48.9** | 38.3 | 36.0 |
| Ours | 47.3 | **47.8** | **47.4** |

Table 11: Classification accuracy (in percentages) of semi-supervised learning results on *tiered*ImageNet. Column headings indicate type of distractor used.

Here, we do not claim that LCBO improves upon semi-supervised learning methods. Especially in the case when distractor inputs are OOS, instead of only OOE examples as studied by Ren et al. (2018), baseline semi-supervised methods significantly degrade the classification accuracy.[3] Yet, since LCBO can detect out-of-distribution examples, it prevents this harmful effect. This

---

[3]This method refers to the baseline formulation of soft $k$-means in Ren et al. (2018).

justifies empirically our motivation that improvements in the out-of-distribution detection can benefit downstream applications.

# H  TEST ACCURACIES

| Model | 5w1s | 5w5s | 10w1s | 10w5s |
|---|---|---|---|---|
| **Protonet** | 53.0 | 70.4 | 40.6 | 57.9 |
| **MAML** | 51.4 | 69.8 | 40.5 | 55.1 |
| **ABML** | 44.8 | 63.7 | 34.6 | 51.5 |

Table 12: Test accuracy for different architectures on CIFAR-100 using Conv4

# I  ADDITIONAL RESULTS FOR CIFAR-100

## I.1  DIFFERENT *shot-/way-* SETTINGS

The overall trend that LCBO and -MinDist are better than SPP holds for different few-shot evaluation settings.

| Metric | AUROC↑ | | | AUPR↑ | | | FPR90↓ | | |
|---|---|---|---|---|---|---|---|---|---|
| **Method** | **SPP** | **-MinDist** | **LCBO** | **SPP** | **-MinDist** | **LCBO** | **SPP** | **-MinDist** | **LCBO** |
| 5w1s OOE | 54.8 | 65.4 | 65.6 | 55.6 | 64.2 | 64.3 | 87.6 | 75.3 | 74.5 |
| 5w1s OOS | 54.6 | 80.5 | 71.6 | 56.8 | 80.3 | 73.1 | 88.3 | 49.2 | 66.8 |
| 5w5s OOE | 60.1 | 68.0 | 73.3 | 61.0 | 67.2 | 71.5 | 84.3 | 73.1 | 62.8 |
| 5w5s OOS | 55.8 | 88.2 | 80.3 | 58.0 | 88.3 | 80.2 | 88.0 | 27.3 | 51.4 |
| 10w1s OOE | 54.0 | 61.1 | 60.7 | 54.9 | 60.2 | 59.8 | 88.6 | 80.1 | 79.9 |
| 10w1s OOS | 56.5 | 80.1 | 62.7 | 58.7 | 79.9 | 66.2 | 86.5 | 47.5 | 84.5 |
| 10w5s OOE | 57.7 | 63.0 | 66.4 | 58.4 | 62.5 | 63.9 | 86.0 | 79.3 | 71.8 |
| 10w5s OOS | 52.0 | 87.9 | 66.2 | 56.6 | 88.0 | 66.0 | 87.8 | 27.7 | 75.0 |

Table 13: Comparison of OOE and OOS detection performance in several way and shot settings for CIFAR-100, using the Conv4 backbone.

## I.2  DIFFERENT "BACKBONE"

Now for better classification accuracies, researchers are moving to larger network architectures, referred to as "backbone" by Chen et al. (2019). Here we show results using the standard Conv4 network and ResNet18 trained with and without data-augmentation (Table 14).

| Metric | AUROC↑ | | | AUPR↑ | | | FPR90↓ | | |
|---|---|---|---|---|---|---|---|---|---|
| **Method** | **SPP** | **-MinDist** | **LCBO** | **SPP** | **-MinDist** | **LCBO** | **SPP** | **-MinDist** | **LCBO** |
| Conv4 OOE | 60.1 | 68.0 | 73.3 | 61.0 | 67.2 | 71.5 | 84.3 | 73.1 | 62.8 |
| Conv4 OOS | 55.8 | 88.2 | 80.3 | 58.0 | 88.3 | 80.2 | 88.0 | 27.3 | 51.4 |
| ResNet18 OOE | 61.1 | 66.1 | 72.2 | 60.3 | 65.0 | 71.6 | 85.4 | 75.2 | 67.5 |
| ResNet18 OOS | 61.4 | 80.4 | 81.1 | 61.2 | 80.5 | 82.3 | 84.8 | 47.3 | 53.1 |
| ResNet18+aug OOE | 61.8 | 73.3 | 77.1 | 58.8 | 72.3 | 74.3 | 82.4 | 65.0 | 54.3 |
| ResNet18+aug OOS | 65.0 | 87.6 | 84.8 | 62.0 | 88.6 | 84.9 | 80.6 | 37.5 | 42.1 |

Table 14: Comparison of the Conv4 and ResNet18 backbones, in the 5-way 5-shot setting.

## J  OTHER RELATED APPROACHES THAT DID NOT MAKE A BIG DIFFERENCE

### J.1  OUTLIER EXPOSURE (HENDRYCKS ET AL., 2019)

We also investigated the effect of outlier exposure (OE) (Hendrycks et al., 2019) for training the LCBO network. We denote LCBO trained with OE as LCBO+OE. Note, however, that this setup differs from that studied by Hendrycks et al. (2019). They do not have a learnable confidence score like LCBO. They simply have a regularization term to encourage the backbone network to output a uniform distribution for OE inputs. We do not train the backbone with OE as they do, but use the OE inputs as additional out-of-distribution examples to train our LCBO network. To train LCBO+OE, we modify the second term in Equation 9 to include queries from the auxiliary dataset, $D$, along with the usual OOE queries $R$:

$$L_{\text{LCBO+OE}}(\phi, \theta; \{S, Q, R\}) = -\sum_{(c, \mathbf{x}^{\text{in}}) \in Q} \log \sigma(\bar{s}_\theta(\mathbf{x}^{\text{in}}, S_c)) - \sum_{\mathbf{x}^{\text{out}} \in R \cup D, c' \sim \text{unif}(V)} \log(1 - \sigma(\bar{s}_\theta(\mathbf{x}^{\text{out}}, S_{c'})))$$

$$(14)$$

The test-time aggregation for LCBO+OE is identical to that described in Section 4.1.

We investigated two auxiliary dataset settings $D$ for LCBO+OE: 1) using the TinyImages dataset as suggested by Hendrycks et al. (2019); and 2) using a combination of TinyImages and the three OOS noise distributions we consider (Gaussian, uniform, and Rademacher noise).

| Metric | AUROC | AUPR | FPR90 |
|---|---|---|---|
| OOE | 72.7 | 70.7 | 63.5 |
| Gaussian | 95.2 | 94.2 | 12.5 |
| Uniform | 74.0 | 72.0 | 62.2 |
| Rademacher | 95.2 | 94.1 | 12.4 |
| Texture | 75.1 | 72.8 | 59.6 |
| Places | 71.1 | 69.9 | 68.4 |
| SVHN | 75.0 | 74.8 | 62.1 |
| LSUN | 71.4 | 71.3 | 69.6 |
| iSUN | 69.4 | 68.5 | 71.3 |
| TinyImagenet | 73.9 | 72.4 | 63.6 |
| OOS MEAN | 77.8 | 76.7 | 53.5 |
| MEAN | 77.3 | 76.1 | 54.5 |

Table 15: Conv4 backbone, 5w5s, LCBO+OE {TinyImages}

| Metric | AUROC | AUPR | FPR90 |
|---|---|---|---|
| OOE | 72.3 | 69.9 | 63.3 |
| Gaussian | 99.9 | 99.9 | 0.2 |
| Uniform | 100.0 | 100.0 | 0.0 |
| Rademacher | 99.9 | 99.9 | 0.2 |
| Texture | 88.9 | 86.0 | 26.8 |
| Places | 72.5 | 70.0 | 61.0 |
| SVHN | 79.3 | 79.9 | 56.5 |
| LSUN | 72.4 | 70.4 | 61.2 |
| iSUN | 74.1 | 71.4 | 57.7 |
| TinyImagenet | 75.8 | 73.2 | 54.8 |
| OOS MEAN | 84.8 | 83.4 | 35.4 |
| MEAN | 83.5 | 82.1 | 38.2 |

Table 16: Conv4 backbone, LCBO + OE {TinyImages, Gaussian, uniform, Rademacher}

### J.2  ODIN

ODIN (Liang et al., 2017) is shown to perform well in the supervised setting. However, as we discussed, SPP does not work well with the Prototypical Network. Below is a Table showing our

attempt to use ODIN in our setting. It slightly improves over SPP, but the improvement is not substantial when compared to -MinDist. We then tried to, like ODIN, perform virtual gradient perturbation. Instead of computing the gradient of the perturbation by probability, we tried perturbing based on the distance in the embedding, so we could improve over -MinDist. However, this approach was not effective in our initial attempts.

| **Metric** | AUROC↑ | | | AUPR↑ | | | FPR90↓ | | |
|---|---|---|---|---|---|---|---|---|---|
| **Method** | **SPP** | **ODIN** | **-MinDist** | **SPP** | **ODIN** | **-MinDist** | **SPP** | **ODIN** | **-MinDist** |
| OOE | 90.2 | 89.8 | 98.6 | 90.0 | 89.9 | 98.8 | 28.3 | 30.4 | 5.2 |
| Gaussian | 17.9 | 20.9 | 100.0 | 35.6 | 37.4 | 100.0 | 94.5 | 94.1 | 0.0 |
| uniform | 85.7 | 88.3 | 100.0 | 90.2 | 92.2 | 100.0 | 37.1 | 31.6 | 0.0 |
| notMNIST | 27.6 | 32.6 | 100.0 | 39.7 | 43.4 | 100.0 | 87.4 | 86.6 | 0.0 |
| cifar10bw | 30.2 | 34.4 | 100.0 | 39.4 | 41.9 | 100.0 | 85.8 | 85.0 | 0.0 |
| MNIST | 16.9 | 20.2 | 100.0 | 35.9 | 38.9 | 100.0 | 95.9 | 94.2 | 0.0 |

