# OpenReview forum: "Out-of-distribution Detection in Few-shot Classification"
_ICLR.cc/2020/Conference — Reject_

### Official Review · AnonReviewer1 · 2019-10-23
**Official Blind Review #1**

**Rating:** 3

**Review:**

Summary

The paper investigates both Out of Distribution (OOS) and Out of Episode (OOE) tasks. The distinction is that OOE samples are from same dataset but come from classes not represented by the support set. The paper shows that existing confidence scores developed in the supervised setting are not suitable when used with popular few-shot classifiers. The paper proposes two new confidence scores, -MinDist and LCBO.

Strengths

The paper proposes benchmark datasets for out-of-distribution detection of few-shot classification. These datasets are Omniglot, CIFAR100, miniImageNet, and tieredImageNet.

The paper presents baseline results for two popular few-shot classifiers — Prototypical Networks, and MAML.

The paper shows that a simple distance metric-based approach improves the performance on both tasks.

It also proposes a parametric, class-conditional confidence score that takes a query x and a class c, and yields a score indicating whether x belongs to class c.

Weaknesses

The paper is very specific to two few shot learning baselines. The question is how representative these two baselines are. Will the distance metric help for other few-shot classifiers?

What is the motivation to detect OOS and OOE in the few shot setting given the accuracy is already low?

The contribution is mainly the metrics.

Overall, the paper does not have enough interesting results for acceptance.


**Experience Assessment:**

I have read many papers in this area.

**Review Assessment: Checking Correctness Of Derivations And Theory:**

I carefully checked the derivations and theory.

**Review Assessment: Checking Correctness Of Experiments:**

I carefully checked the experiments.

**Review Assessment: Thoroughness In Paper Reading:**

I read the paper thoroughly.

---

> ### Author Response · Authors · 2019-11-14
> **motivation, relevance [1/2]**
>
> We thank you for taking the time to read and understand our paper. The two weaknesses raised are:  1) that ProtoNet and MAML are not representative enough of all few-shot classifiers, and 2) that few-shot classifiers are not accurate now, so OOD detection in general should not be pursued.
>
> We believe that the two concerns are related. Few-shot classification, by definition, is a harder problem than supervised classification due to the limited amount of data. This is not to say that we have reached the ceiling. In fact, few-shot learning is a growing research topic in our community. As discussed in a review on few-shot learning (refer to Section 2 of [1]), most few-shot algorithms fall into "Distance learning", "Learning to fine-tune", and other add-on methods.
>
> We believe that ProtoNet and MatchingNet [2] are the most representative of the first class. Studies have shown that ProtoNet often outperforms MatchingNet [3, 1, 4], hence we chose ProtoNet as our representative of the first class. MAML on the other hand, is often taken as a representative method of the second class in review studies [1, 4], because other "learning to fine-tune" methods are often discussed as extensions to MAML.
>
> Precisely because of its low accuracy, OOD detection can be useful for semi-supervised learning with distractors [5], active learning [6] to practically improve the classifier, or at least defer the decision when being deployed in real-world situations, such as using few-shot learning to model the long-tail distribution of rare medical conditions  [7]. Our contribution is, indeed, mainly about the new benchmark, and providing a metric for the few-shot community. An increasing number of studies deal with building new few-shot classifiers with better uncertainty estimates [8-11], but suffer from a lack of a good metric, which underscores the relevance of a study like ours.
> References are given in part 2 of this comment, below.

---

> > ### Author Response · Authors · 2019-11-14
> > **motivation, relevance [2/2]**
> >
> > REFERENCES
> > [1] Wei-Yu Chen, Yen-Cheng Liu, Zsolt Kira, Yu-Chiang Wang, and Jia-Bin Huang. A closer look at few-shot classification.  In International Conference on Learning Representations (ICLR), 2019.
> > [2] Oriol Vinyals, Charles Blundell, Tim Lillicrap, Daan Wierstra, et al. Matching networks for one shot learning. In Advances in Neural Information Processing Systems (NIPS), 2016.
> > [3] Jake Snell, Kevin Swersky, and Richard Zemel. Prototypical networks for few-shot learning. In Advances in Neural Information Processing Systems (NIPS), pages 4077–4087, 2017.
> > [4]  Eleni Triantafillou, Tyler Zhu, Vincent Dumoulin, Pascal Lamblin, Kelvin Xu, Ross Goroshin, Carles Gelada, Kevin Swersky, Pierre-Antoine Manzagol, and Hugo Larochelle. Meta-dataset: A dataset of datasets for learning to learn from few examples. arXiv preprint arXiv:1903.03096, 2019.
> > [5] Mengye Ren, Eleni Triantafillou, Sachin Ravi, Jake Snell, Kevin Swersky, Joshua B Tenenbaum, Hugo Larochelle, and Richard S Zemel.  Meta-learning for semi-supervised few-shot classification. arXiv preprint arXiv:1803.00676, 2018.
> > [6] Yarin Gal, Riashat Islam, and Zoubin Ghahramani. Deep Bayesian active learning with image data. In International Conference on Machine Learning (ICML), pages 1183–1192, 2017.
> > [7] Viraj Prabhu, Anitha Kannan, Murali Ravuri, Manish Chablani, David Sontag, and Xavier Amatriain. Prototypical clustering networks for dermatological disease diagnosis. arXiv preprint arXiv:1811.03066, 2018.
> > [8] Sachin Ravi and Alex Beatson. Amortized Bayesian meta-learning. In International Conference on Learning Representations (ICLR), 2019.
> > [9] Tyler R Scott, Karl Ridgeway, and Michael C Mozer. Stochastic prototype embeddings. arXiv preprint arXiv:1909.11702, 2019.
> > [10] Massimiliano Patacchiola, Jack Turner, Elliot J Crowley, and Amos Storkey.  Deep kernel transfer in Gaussian processes for few-shot learning. arXiv preprint arXiv:1910.05199, 2019.
> > [11] Vincent Dutordoir, Hugh Salimbeni, James Hensman, and Marc Deisenroth. Gaussian process conditional density estimation. In Advances in Neural Information Processing Systems (NIPS), pages 2385–2395, 2018.

---

### Official Review · AnonReviewer2 · 2019-10-24
**Official Blind Review #2**

**Rating:** 3

**Review:**

This work investigates a new problem setting that combines few-shot image classification and out-of-distribution detection. The main procedure for the task still follows a standard few-shot classification task, but in each episode, the data from a different distribution may be presented together with the query images. The evaluation focuses on the performance of out-of-distribution detection, which relies on a scoring function to assign a high score for the normal query images while having a low score for the out-of-distribution images. The paper evaluates with three different scoring functions on four few-shot classification datasets and nine out-of-distribution datasets.

We recommend a weak rejection, although the proposed new problem setting might be interesting to the community, due to three major concerns. The first is insufficient description for reproducing the evaluation. For example, the text mentions “All results are evaluated using 1000 test episodes” without information of how the in-distribution data and out-of-distribution data (especially the OOS data) is chosen in each episode. Second, the experiments and discussions do not provide enough insights for readers to understand the impact of combining the two problems. Some questions should be addressed and are listed below. The last is the writing style which has a noticeable fraction of content not directly related to the proposed problem setting. Such a style can confuse readers and require additional passes of reading to understand.

Some questions to be addressed for the second concern:
1. What’s the impact of doing out-of-distribution under the few-shot setting? Is it harder than a normal out-of-distribution detection setting? How does the “N-way X-shot” setting affect the difficulty of the problem?
2. The paper proposes to use -MinDist and LCBO for the scoring functions instead of the methods that are commonly seen in a standard out-of-distribution detection paper. Why not use those previous methods (ex: ODIN, Mahalanobis, ensemble strategies, etc.) for the evaluation? If those previous methods do not work well with the proposed new setting, what are the possible causes?

Examples of the third concern include: (1) The MAML in Section 2 has a whole paragraph that could be summarized in a few sentences. (2) Figure 3 draws the schematics of out-of-distribution detection, but its connection to the proposed setting is not clearly described. (3) The introduction mentioned semi-supervised learning and continual learning, which does not strengthen the argument of why few-shot learning and out-of-distribution detection should be combined. (4) Lastly, the FS-SSL paragraph in the experiment section has no conclusion.


**Experience Assessment:**

I have published one or two papers in this area.

**Review Assessment: Checking Correctness Of Derivations And Theory:**

I carefully checked the derivations and theory.

**Review Assessment: Checking Correctness Of Experiments:**

I carefully checked the experiments.

**Review Assessment: Thoroughness In Paper Reading:**

I read the paper thoroughly.

---

> ### Author Response · Authors · 2019-11-14
> **Clarification, added description**
>
> We thank you for taking the time to read and understand our paper. We agree with many of the concerns you raised. Some were solid feedbacks, which we incorporated in our revision to improve our paper. Some others were actually addressed in the original write-up. We hope to provide you with pointers to them in this comment. As you raised 3 main concerns, we will respond to each here.
>
> 1. "Insufficient description for reproducing evaluation .... specifically how OOS are selected".
>
> Thank you for this point. We addressed it as follows:
> a. We updated Appendix B to include a description of episodic evaluation.
> b. We attached one version of our code in case you’re interested in other reproducibility details. Also, we plan to refine and open-source our code in the near future.
>
> 2. "Discussion of results"
> 2.1. "Is Few-Shot OOD harder than standard OOD?", "What is the effect of varying the way/shot setting?"
>
> Due to a lack of space, we only mentioned this analysis in the first paragraph of Section 5.2, and further results and discussion were included in Appendix I. In brief, this speaks to why OOD in the few-shot setting is interesting. The more challenging the few-shot classification task is, the more challenging the OOD task becomes as well. As we increase the way, the task becomes harder, whereas increasing the shot makes the task easier.
>
> 2.2. "Why not use those previous methods (ex: ODIN, Mahalanobis, ensemble strategies, etc.)".
>
> These points are explicitly discussed in both our main text and the appendix, if you could please direct your attention.
> With regard to ODIN, we state that: "Our preliminary experiments showed that ODIN did not have a big impact in the few-shot setting. ‘Outlier exposure’ another recent method (Hendrycks et al., 2019) also did not show a significant effect." We included results in Appendix J.
> With regard to Mahalanobis, we discuss it extensively in Section 5.2 and Appendix E.
> We would appreciate it if you could revisit these points.
>
> 3.  "Writing style"
>
> Our motivation for this paper, as you correctly noted, is to "investigate a new problem setting that combines few-shot image classification and out-of-distribution detection." In this case, we do not expect our readers to be experts in both topics, or either. Hence, we specifically chose a writing style where we can provide a good enough starting point for readers to get started on these topics, with relevance to this newly proposed task.
>
> Lastly, you said that our FS-SSL section has no conclusion, which is not the case because we stated in our paper: "Yet, especially in the case when distractor inputs are OOS, instead of only OOE examples, baseline semi-supervised methods significantly degrade the classification accuracy." This speaks to exactly why one would care about OOD detection in this setting. However, we are open to making it more explicit.

---

### Official Review · AnonReviewer3 · 2019-11-01
**Official Blind Review #3**

**Rating:** 3

**Review:**

This paper presents a method for out-of-distribution (OOD) sample detection for cases that only a few samples are available from positive classes. The authors bridge the gap between novelty detection and few-shot classification models. The paper addresses an interesting task in machine learning but in point of view of the out-of-distribution detection task, the proposed method seems to be an incremental task and far from the state-of-the-art.

--The authors claim that the combination of one-class classification and few-shot classification has not been studied before, while this task is implicitly mentioned in previous work. For instance, these topics are explored in “Out-of-Distribution Detection for Generalized Zero-Shot Action Recognition” for a specific task.
--The authors divided the out-of-dataset (OOS) into three categories, while the most famous state-of-the-art methods for one class classification task especially those that are devised based on reconstruction error or adversarial training schemes do not fall in any of these categorized.
--Detecting the OOS samples task is very similar to recognizing the membership attack task, but the authors do not mention anything about them.
--The proposed model has access to the negative samples (OOD samples) during training. First, this assumption is not realistic, as in all such applications, the irregular (OOD) samples are either poorly sampled or not sampled at all. Second, most of the previous works on OOD assume that OOD samples are not available during training. This makes them not directly comparable to the proposed methods that its settings are more-or-less advantaged.
--Page 4 – what is D_c’ or C’ in Eqs. (3) and (4)?



**Experience Assessment:**

I have published in this field for several years.

**Review Assessment: Checking Correctness Of Derivations And Theory:**

I carefully checked the derivations and theory.

**Review Assessment: Checking Correctness Of Experiments:**

I assessed the sensibility of the experiments.

**Review Assessment: Thoroughness In Paper Reading:**

I read the paper at least twice and used my best judgement in assessing the paper.

---

> ### Author Response · Authors · 2019-11-14
> **Added discussion**
>
> We thank Reviewer 3 for pointing us to the related paper ‘Out-of-Distribution Detection for Generalized Zero-Shot Action Recognition’ [1]. We find this work interesting and we have updated our Related Work section to mention it.
>
> However, we would like to point out that the setting in [1] is different: at inference time, their model is given access to category-specific embeddings of the unseen classes. These are then used to condition a generative model in order to generate features from the unseen classes. Their OOD detector is then trained to discriminate between seen and unseen features, by considering a supervised learning problem treating the generated data as the negative class. Specifically, the way their OOD detector is trained is very similar as used in both [2],  and Outlier Exposure (OE) [3] , all encouraging a uniform prediction on surrogate outliers (e.g.  generated data or otherwise prepared datasets).   We actually included results on using OE (in Related work and Appendix J), which is not as effective as our proposed method in our case. We think this is exactly because of the difference in setting.
>
> In our setup, we encounter new classes at inference time, given only a small handful of examples from each class (the support set). Our goal is to reliably understand the class boundaries of these new classes, given only that small support set. The task then is to correctly classify other query examples into one of the new classes, while correctly identifying if a query example is OOD (that is, does not belong to any of the classes in the support set). Importantly, the OOD samples are encountered only in the query set, so at the time of learning class boundaries of an episode’s new classes (using the support set), we don’t have access to any OOD samples or to any OOD class labels and thus to any way of generating positive instances of OOD classes. We therefore cannot train a binary classifier to decide if an example is OOD or not. Ours is a challenging and interesting setup, and as far as we know, we are the first to propose to study it.
>
> Regarding the note that ‘The proposed model has access to the negative samples (OOD samples)during training’ – this assertion is in fact not correct! At training time we only use samples of the training classes. To form each episode, as we have described in the paper, we first sample a set of classes to form its in-distribution set of classes, and then we sample some different classes (from the training set as well!) to act as its out-of-distribution set of classes. The latter only participate in the query set. This idea is akin to how we use support and query sets to structurally play the role of training and testing sets for each episode, even though at training time both support and query are populated by training examples.
>
> REFERENCES
> [1] Devraj  Mandal,  Sanath  Narayan,  Sai  Kumar  Dwivedi,  Vikram  Gupta,  Shuaib  Ahmed,  Fa-had Shahbaz Khan, and Ling Shao. Out-of-distribution detection for generalized zero-shot action recognition.  In Conference on Computer Vision and Pattern Recognition, pages 9985–9993, 2019
> [2] Lee, Kimin, et al. "Training confidence-calibrated classifiers for detecting out-of-distribution samples." arXiv preprint arXiv:1711.09325 (2017).
> [3] Dan Hendrycks, Mantas Mazeika, and Thomas Dietterich. Deep anomaly detection with outlier
> exposure. In International Conference on Learning Representations (ICLR), 2019. URL https:
> //openreview.net/forum?id=HyxCxhRcY7

---

### Author Response · Authors · 2019-11-14
**responses and revisions**

We thank all the reviewers and the AC for taking the time to review our work. We found many of the comments very helpful, and by incorporating them into our revision we have improved our write-up.

We have addressed the concerns in individual comments. In terms of revisions, we:
1. Added a description of OOD evaluation, and attached a preliminary version of publishable code.
2. Added a discussion of related work as pointed out by R3.

---

### Decision · Program_Chairs · 2019-12-19

**Decision:**

Reject

**Comment:**

This paper presents a method for out-of-distribution detection under the condition of access to only a few positive labeled samples. The main contribution as summarized by reviewers and authors is the new proposed benchmark and problem statement.

All reviewers are in agreement that this paper is not ready for publication in its current form. The main concern is around the validity of the problem statement. The reviewers seek more clarity motivating the proposed scenario. Though the authors argue that as few-shot recognition is very difficult and may benefit from strategies like active learning, it is not directly clear how out of distribution detection is the best approach. In addition, R3 seeks clarification on the similarity to existing work.

Considering the unanimous opinions of the reviewers and all author rebuttal text, the AC does not recommend acceptance of this work. We encourage the authors to focus their revisions on the explanation and motivation of this new benchmark and submit to a future venue.